# Understanding and Exploiting Post-Translational Modifications for Plant Disease Resistance

**DOI:** 10.3390/biom11081122

**Published:** 2021-07-30

**Authors:** Catherine Gough, Ari Sadanandom

**Affiliations:** Department of Biosciences, Durham University, Stockton Road, Durham DH1 3LE, UK; catherine.gough@durham.ac.uk

**Keywords:** post-translational modifications, plant immunity, phosphorylation, ubiquitination, SUMOylation, defence

## Abstract

Plants are constantly threatened by pathogens, so have evolved complex defence signalling networks to overcome pathogen attacks. Post-translational modifications (PTMs) are fundamental to plant immunity, allowing rapid and dynamic responses at the appropriate time. PTM regulation is essential; pathogen effectors often disrupt PTMs in an attempt to evade immune responses. Here, we cover the mechanisms of disease resistance to pathogens, and how growth is balanced with defence, with a focus on the essential roles of PTMs. Alteration of defence-related PTMs has the potential to fine-tune molecular interactions to produce disease-resistant crops, without trade-offs in growth and fitness.

## 1. Introduction

Plant growth and survival are constantly threatened by biotic stress, including plant pathogens consisting of viruses, bacteria, fungi, and chromista. In the context of agriculture, crop yield losses due to pathogens are estimated to be around 20% worldwide in staple crops [1]. The spread of pests and diseases into new environments is increasing: more extreme weather events associated with climate change create favourable environments for food- and water-borne pathogens [2,3].

The significant estimates of crop losses from pathogens highlight the need to develop crops with disease-resistance traits against current and emerging pathogens. Crop protection methods have low effectiveness against pathogens, which includes fungicides and insecticides which control insect viral transmissions; moreover, resistance against these chemicals is increasing [4,5]. Resistance refers to the inability of a pathogen to complete its life cycle on that plant species [6]; targeting host resistance for improvement is the most economical and effective method for controlling the reduction in crop losses to disease [7,8,9].

Developing novel solutions to this increasing problem requires a deeper understanding of plant defence mechanisms. Beyond gene expression and transcriptomics, proteomics is particularly useful as it can directly measure relative protein abundance, as well as detecting post-translational modifications (PTMs) [10]. PTMs can activate, deactivate, or change protein function to induce or attenuate specific plant responses. Analysis at the protein level can reveal pathogen host targets, protein turnover, and protein–protein interactions in defence signalling for the purpose of modifying and boosting immunity in crops [11]. This review outlines the functions of PTMs in immunity and the potential to manipulate PTMs to enhance disease resistance.

## 2. The Framework of Plant Defence

Due to their sessile nature, plants rely heavily on chemical defences against biotic and abiotic stresses [11]. Plants are constantly challenged by biotic stresses: pathogen infection harms plant growth, reproduction, and survival. Plants have defence systems to overcome or reduce pathogen attacks, which include physical barriers to prevent pathogen entry and an innate immune system to respond to pathogen attacks [12].

The inducible plant innate immune system is made up of PAMP-triggered immunity (PTI) and effector-triggered immunity (ETI) which significantly overlap (Figure 1) [13,14,15,16]. Arabidopsis (*Arabidopsis thaliana*) is used as the predominant model system to study the molecular events of the defence signalling pathway, but the overall system is conserved in monocots and dicots [17]. Conserved molecular structures from microbes known as pathogen-associated molecular patterns (PAMPs) are recognised by cell-surface pattern recognition receptors (PRRs), triggering downstream immune responses [13,18,19]. Well-defined PAMP–PRR interactions are bacterial flagellin peptide flg22 and its cognate receptor FLAGELLING-SENSING 2 (FLS2) [20,21], bacterial elongation factor thermal unstable (EF-Tu) and its receptor EF-Tu receptor (EFR) [22,23], and fungal cell wall polysaccharide chitin and its receptor CHITIN ELICITOR RECEPTOR KINASE 1 (CERK1) [24,25]. These PRRs are receptor-like kinases which initiate a phosphorylation cascade to activate defence-related enzymes or genes [26] (Figure 1).

Several effectors are secreted by the pathogen to disrupt cellular function throughout infection. Unlike PAMPs, effectors are diverse and include proteins, sRNAs, chemicals, toxins, and hormones which increase pathogen infection by benefiting the pathogen or suppressing host defence. Intracellular receptors called nucleotide-binding domain, leucine-rich repeat-containing proteins (NLRs, also known as NB-LRRs) detect specific effectors delivered into the plant cell to trigger effector-triggered immunity (ETI). NLRs can themselves detect effectors or function as helpers to trigger signal transduction [27]. Detection by NLRs is either directly (receptor-ligand model) or, in most cases, indirectly through the ‘guard’ or ‘decoy’ mechanisms [28,29]. The two major groups of NLRs are Toll-interleukin-1 receptor-like nucleotide-binding site leucine-rich repeat (TNL) and coiled-coil (CC)-NBS-LRR (CNL) [27,30]. Furthermore, resistance to powdery mildew 8 (RPW8)-NBS-LRRs (RNLs) function as helper NLRs [31,32]. Proteins downstream of NLR activation include NON-RACE-SPECIFIC DISEASE RESISTANCE 1 (NDR1) and ENHANCED DISEASE SUSCEPTIBILITY 1 (EDS1) [33] (Figure 1). These pathways lead to outcomes including accumulation of salicylic acid and defence gene activation [34,35].

Molecular and physiological changes induced downstream of PRRs and NLRs include mitogen-activated protein kinase (MAPK) activation, production of reactive oxygen species (ROS), stomata closure, defence gene expression, hypersensitive response (HR) and cell death, callose deposition and lignification, reduction of photosynthesis, increased respiration and expression of PATHOGENESIS-RELATED (PR) proteins, and production of antimicrobial compounds [10,11,36,37,38]. Pathogen perception triggers changes in hormone levels including salicylic acid (SA) which mediates defences against biotrophs and hemi-biotrophs and jasmonic acid (JA)/ethylene which mediates defences against necrotrophs [39].

The understanding of the plant defence response in the model organism *Arabidopsis* or in crops is not complete. Many studies have relied on genomic or transcriptomics; however, transcriptional changes do not reflect the complete cellular regulatory processes, since post-transcriptional processes that alter the amount of active protein, synthesis, degradation, processing, and modification of proteins are not taken into account. Thus, complementary approaches such as proteome-based expression profiling are needed to obtain a full picture of the regulatory elements in plant–pathogen interactions [40]. At almost every stage of defence, PTMs are important, allowing fast activation and signalling; PTMs act as molecular switches to alter protein functions rapidly [41,42]. This review considers PTMs in crop improvement, among other approaches. PTMs modification may offer a more nuanced approach and provoke less of a yield penalty than gene knockouts or gene introduction.

## 3. Post-Translational Modifications Have a Critical Role in Defence

Post-translational modifications are critical for plant defence responses and are involved in almost every aspect of plant growth and development. PTMs allow protein function to be extended above that of its structure determined by the primary amino acid sequence to control almost all characteristics of protein function. PTM systems are targeted by numerous pathogen effectors; thus, PTMs are worthwhile to investigate in terms of modification and exploitation in crops. This work will focus on phosphorylation, ubiquitination, and SUMOylation, the most well-studied PTMs, which are reversible (Figure 2). Others to briefly mention are N-myristoylation, S-acylation, S-nitrosylation, acetylation, glycosylation, sulphenylation, and redox modification, which also have roles in immunity [42,43] but are not covered in this review. Reversibility is crucial to regulate the intensity and duration of protein activity and defence response [44].

### 3.1. Phosphorylation

Phosphorylation is paramount in several aspects of immunity to control enzyme activity and in signalling. Phosphorylation is crucial in PRR downstream responses through phosphorylation cascades; phosphorylation is a rapid and transient switch (Figure 2) and is essential in immune signal transduction [42]. Ligand perception in several PRRs stimulates the recruitment of coreceptor BRI1-ASSOCIATED RECEPTOR KINASE (BAK1) (also known as SOMATIC EMBRYOGENESIS RECEPTOR KINASE 3 (SERK3)), which heterodimerises with several receptor-like-kinases (RLKs) including FLS2, BRASSINOSTEROID INSENSITIVE-1 (BRI1), and EFR [45]. BAK1 is differentially phosphorylated when in complex with different PRR complexes [46]. BOTRYTIS-INDUCED KINASE 1 (BIK1) is a substrate of BAK1 and the pair feature in many defence signalling pathways. Autophosphorylation and transphosphorylation are essential by both BIK1 and BAK1 in their interaction and interaction with other downstream components in signal transmission [45]. BIK1 dissociation activates downstream signalling, such as activation of MAPK cascades, transcriptional reprogramming, and ROS production [47,48]. BIK1 directly phosphorylates respiratory burst oxidase homologue protein D (RbohD), an NADPH oxidase which produces ROS burst to induce stomatal closure and act as antimicrobial molecules [49,50].

BAK1 is a key kinase in plant immunity which possesses numerous phosphorylation sites itself to regulate specific outputs, as shown by mutagenesis studies. Some phosphorylation sites have positive effects, and some have negative effects on BAK1 function [51,52]. The T455A (threonine-to-alanine) mutation abolishes BAK1 kinase activity, and the conserved BAK1 residue Y403 is important for ligand-induced activation of the immune receptor complex [53]. Phosphorylation patterns are specific to mediate the response allowing BAK1 to regulate defence and brassinosteroid signalling. For example, it was suggested that specific BAK1 mutant variants BAK1^C408Y^ and BAK1^T450A^ provoke differential phosphorylation patterns on specific receptors. This conclusion was drawn as the mutant phenotypes BAK1^C408Y^ and BAK1^T450A^ show impaired defence signalling but with wild-type (WT)-like BAK1-mediated brassinosteroid (BR) signalling [53,54]. These phenotypes are different from the BAK1 null allele; thus, clear mutations in specific residues can alter phenotype [46,55]. Interestingly, the mutation in C408 reduced phosphorylation of Y403, shown using a specific pY403 antibody, which highlights that residues surrounding the PTM attachment site can influence PTM status [56]. This mutagenesis approach to potentially mutate residues surrounding the PTM could be advantageous to stabilise/destabilise PTMs without blocking the PTM formation, to make interactions reduced or enhanced in some circumstances.

It is clear that phosphorylation is central to defence in signal transduction [57]; the activation of the MAPKs MPK3, MPK4, and MPK6 (MPK3/4/6) is a hallmark of immune system activation and is crucial for establishing disease resistance [58]. All known PRRs activate two MAPK cascades (Figure 1) consisting of MAPK kinase kinase (MKKK), MAPK kinase (MKK), and MAPKs: MAPKKK3/MAPKKK5-MKK4/MKK5-MPK3/MPK6 which positively regulates defence, and MEKK1 - MKK1/MKK2-MPK4 which negatively regulates immune responses [58,59,60,61,62]. Phosphorylation of downstream substrates such as WRKY transcription factors causes transcriptional changes [63]. For instance, WRKY33 is a substrate of MPK3/6 which activates transcription of *PHYTOALEXIN DEFICIENT 3 (PAD3)* encoding a cytochrome P450 enzyme (CYP71B15) which carries out the last step of camalexin biosynthesis, causing induction of camalexin, which has antimicrobial effects [63,64]. Additionally, MPK3/6 activation is critical to including inhibition of photosynthesis to promote ROS accumulation in chloroplasts and HR cell death [65]. Moreover, MPK4 is targeted by *Pseudomonas syringae* bacterial type III effector HopAI1 and acts as the guardee of NLR SUPPRESSOR OF MKK1 MKK2 2 (SUMM2) [66]. Disruption of the MEKK1-MKK1/2-MPK4 kinase cascade results in constitutive immune responses mediated by the NLR protein SUMM2 [67].

Reversibility is paramount to control phosphorylation states to regulate signal transduction, constitutive activation of defence leads to growth defects [68]. Phosphorylation of PRR complexes including FLS2-BAK1-BIK1 is negatively regulated by PROTEIN PHOSPHATASE TYPE 2A (PP2A) and PROTEIN PHOSPHATASE TYPE 2C (PP2C) [42,69,70]. Similarly, CERK1-INTERACTING PROTEIN PHOSPHATASE 1 (CIPP1) dephosphorylates CERK1, in the absence of chitin, to negatively regulate CERK1 signalling [71]. Phosphatases ARABIDOPSIS PHOSPHATASE 2Cs (AP2Cs) interact with MPK3, 4, and 6 to negatively regulate innate immunity against necrotrophic fungal pathogen *Botrytis cinerea* [72,73]. MAP KINASE PHOSPHATASE1 (MKP1) and PROTEIN TYROSINE PHOSPHATASE1 (PTP1) act as repressors of inappropriate MPK3/MPK6-dependent stress signalling [74,75]. Additionally, phosphorylation can lead to feedback dephosphorylation; for example, MKP1 is phosphorylated by MPK6, one of MKP1’s substrates [76].

### 3.2. Ubiquitination

Ubiquitin (Ub) is covalently attached to specific lysine residues of target proteins through an enzymatic cascade, which is reversible (Figure 2) [77]. Most ubiquitylated proteins, especially those modified with lysine48(K48)-linked polyubiquitin chains, are targeted for degradation by the 26S proteasome [78,79]. Nevertheless, ubiquitination has several functions including signalling, endocytic trafficking, etc., dependent on the specific attachment linkage [80,81]. The ubiquitin system is required for innate immunity and its regulation [82,83]. For example, expression of a ubiquitin variant with a K48R (lysine-to-arginine) change prevents K48 attachments (Figure 2) and alters the responses to viruses in tobacco [82]. K48 is one of the most abundant ubiquitin attachments that cause ubiquitin-mediated proteasomal degradation, although other linkages may be involved [77,84]. Different enzymes of ubiquitin machinery impact immunity. *Arabidopsis* has two Ub E1s, UBIQUITIN ACTIVATING ENZYME 1 (UBA1) and UBA2, which are partially redundant. The null mutant of UBA1, *mos1*, is defective in innate immunity, whereas the *uba2* null mutant plants do not have defects in immunity. It was shown that the activation and downstream signalling of several resistance (R) proteins requires Ub E1 UBA1 [83].

Many E3 ubiquitin ligases are involved in plant immunity by carrying out ubiquitination to target substrates [85]. Ubiquitination is essential to regulate levels of plant immune system components through protein turnover to avoid excessive or inappropriate responses. This is illustrated by the *plant u-box 13 (pub13)* mutant which has enhanced immune responses on pathogen attack or flg22 perception. However, the *pub13* mutant demonstrates autoimmune responses, namely, causing spontaneous cell death and accumulation of ROS in the absence of stress, which shows the importance of PTM regulation [86]. It was further shown that FLS2 is specifically polyubiquitinated by ubiquitin E3 ligases PUB12/13 which target FLS2 for degradation. Interestingly, phosphorylation by BAK1 activates PUB12/13 after FLS2 binds flagellin, demonstrating the feedback attenuation of FLS2 responses and the reliance on multiple PTMs in defence regulation [86]. BAK1 kinase activity is essential in mediating its interaction with PUB13 since the BAK1 kinase-inactive mutant which has the K317M substitution could no longer interact with PUB13 [87]. PUB13 also ubiquitinates LYSM-CONTAINING RECEPTOR-LIKE KINASE 5 (LYK5), targeting it for degradation to regulate chitin-triggered defences (Figure 1) [88]. Ub E3 ligase PUB25/26 targets nonactivated immune kinase BIK1 for degradation to modulate BIK1 levels (Figure 1) [89]. PUB4 interacts with CERK1 and is a positive regulator of chitin-induced immune responses [90].

Overaccumulation of NLRs often leads to autoimmune responses. In order to prevent this, NLR proteins SUPPRESSOR OF NPR1-1, CONSTITUTIVE 1 (SNC1), and RESISTANT TO P. SYRINGAE 2 (RPS2) are targeted for ubiquitination and degradation by the SKP1-CULLIN1-F-box (SCF) complex (Cheng et al., 2011). In contrast, *Arabidopsis* Ub E3 ligases RPM1 INTERACTING PROTEIN 2 and 3 (RIN2 and RIN3) contribute positively to NLRs RESISTANCE TO P. SYRINGAE PV. MACULICOLA 1 (RPM1)- and RPS2-dependent HR (Kawasaki et al., 2005).

Ubiquitination is essential to allow activation of JA responses against necrotrophs. JASMONATE-ZIM-DOMAIN PROTEIN 1 (JAZ) proteins function as transcriptional repressors of JA-responsive genes [91]. Bioactive JA (jasmonoyl–isoleucine (JA–Ile) conjugate) promotes the physical interaction between the ubiquitin ligase complex SCF^COI1^ and JAZ proteins to cause ubiquitin-mediated proteasomal degradation of JAZ, to allow expression of JA-dependent genes [91,92,93].

Although Ub E3s largely determine the substrate specificity [94,95,96,97], deubiquitinating enzymes (DUBs) also have substrate specificity [80,98]. This is important in immunity; for example, deubiquitinating enzymes *Arabidopsis* UBIQUITIN-SPECIFIC PROTEASE 12 and 13 (AtUBP12 and AtUBP13) were found to negatively regulate plant immunity [99]. However, UBP12 and UBP13 are positive regulators of JA responses and may act by stabilising MYC, resulting in the JA pathway suppressing SA-mediated immunity [100].

### 3.3. SUMOylation

Besides ubiquitin, ubiquitin-like polypeptides are covalently conjugated to substrates in eukaryotes via the substrate lysine (Figure 2). Small ubiquitin-like modifier (SUMO) is another important PTM involved in plant biotic stress responses. Global SUMOylome changes occur on pathogen attacks [101,102,103,104]. For example, the auto-immune *suppressor of rps4-rld1-4 (srfr1-4)* mutants showed large increases in basal SUMO1/2-conjugates, as did wild-type plants challenged with *Pseudomonas Syringae pv. tomato (Pst)DC3000*, compared to the WT untreated plants. Overall, the *srfr1-4* mutant and *PstDC3000* infected WT plants were found to share 57.9% of their common SUMO substrates which consist of wide-ranging targets. The autoimmune *srfr1-4* plants have increased SA levels and constitutive upregulation of PR1/PR2 genes; stunted growth is also observed [105]. Significantly, loss of EDS1 restores the SUMOylome in *srfr1-4* to wild-type (Col-0) levels and abolishes growth retardation and autoimmunity [106]. Therefore, SUMOylation and deSUMOylation are crucial for defence regulation.

Different SUMO paralogues have different functions, and different paralogs exist in different species [107]. In *Arabidopsis*, SUMO1/2 inhibits SA-mediated defence responses in the absence of pathogen [108]. In contrast, SUMO3 promotes plant defence responses downstream of SA [109]. SUMO also forms noncovalent interactions with proteins via SUMO interacting motifs (SIMs) which facilitate interactions between SUMO-conjugated proteins and protein partners featuring SIM site(s) for protein complex formation [107,110].

Altering the specific pattern of PTMs changes the plant’s defence responses and ability to resist disease. For instance, the OVERLY TOLERANT TO SALT1 and -2 (OTS1/2) SUMO protease double mutant *ots1ots2* accumulates increased levels of SUMO conjugates, higher levels of SA, and enhanced resistance to *PstDC3000*, compared to WT plants. It was found that SUMO proteases OTS1 and OTS2 limit SA biosynthesis by suppressing *ISOCHORISMATE SYNTHASE1 (ICS1)* expression, and as a feedback mechanism, SA promotes the degradation of OTS1 and OTS2 in order to modulate SA signalling [101]. Similarly, SUMO protease mutants *early in short days 4 (esd4)* have high SA accumulation [111]. These show that the SUMO enzymatic machinery regulates SA-mediated defence to adjust the response appropriately [101,109].

Another aspect of SUMO machinery to affect defence is portrayed by the *sap and miz 1* (*siz1)* SUMO E3 ligase *Arabidopsis* loss-of-function mutant. *siz1* plants have decreased SUMO conjugates, dwarfism, an autoimmune phenotype, characterised by increased accumulation of SA, increased expression of *EDS1*, *PAD4,* and *PATHOGENESIS-RELATED* (*PR)* genes, and greater resistance to the bacteria *PstDC3000*, compared to WT plants [112]. The *siz1* autoimmune phenotype is dependent on the TNL immune receptor SNC1 [113,114]. TOPLESS-RELATED 1 (TPR1), an SNC1-interacting protein, physically interacts with and is SUMOylated by SIZ1 [115]. Mutation of K282 and K721, the critical SUMOylation attachment sites of TOPLESS-RELATED 1 (TPR1) suggested that TPR1 SUMOylation represses immunity through repression of its own transcriptional co-repressor activity. This leads to the expression of negative regulators of immunity *DEFENSE NO DEATH 1* (*DND1*) and *DND2*. In addition, SNC1 is SUMOylated, which perhaps further acts to repress immunity in the absence of pathogens [113,115]. *SNC1* transcription is controlled by SUMOylation, as well as SNC1 being SUMOylated at the protein level [113], and SNC1 protein level is controlled by ubiquitin-mediated degradation, as mentioned in the previous section [116]. It is important to control SNC1 activity to avoid excessive immune responses, which would be detrimental to plant growth and cause damage [117].

Disruption of the PTM enzymatic machinery underlines the fact that changes in PTM attachment/removal have profound effects on plant physiology, including regulation of defence.

Interestingly, increased SUMOylation in *ots1ots2* mutants or reduced SUMOylation *siz1* mutants have increased SA levels indicating the complexity in PTM regulation, and that SUMOylation regulation is key to modulate the correct level of immunity. This tight control of SUMO is further highlighted as overexpression of the three Arabidopsis *SUMO (SUM)* genes resulted in activation of SA-dependent defence responses, as did the *sum1sum2* knockdown mutant [109].

In addition to SA signalling, SUMO has a role in modulating JA signalling. SUMO-conjugated to JAZ inhibits the JA Receptor CORONATINE INSENSITIVE1 (COI1) through the COI1 SIM site [118]. SUMO protease OTS1/2 action or degradation determines if the JA response is activated or inhibited, dependent on the type of pathogen [118]. Significantly, SUMOylation interacts with other PTMs including phosphorylation and ubiquitination which will be outlined in the subsequent section.

### 3.4. Interaction between PTMs

Most aspects of immunity are regulated by multiple PTMs, which often interact. PTMs undergo crosstalk and have a mutual dependence. One prominent example is FLS2 signalling whose regulation requires phosphorylation, SUMOylation, and ubiquitination [48,86,119]. In uninfected conditions, FLS2 associates with BIK1 [21,120]. When flg22 is detected, FLS2 recruits coreceptor protein kinase BAK1 which allows BIK1 and BAK1 to undergo reciprocal phosphorylation [55,121,122]. Additionally, on flagellin perception, FLS2 is SUMOylated on lysine1120, triggering the release of BIK1, which is essential for the FLS2-mediated defence response. DeSUMOylating isopeptidase 3A (Desi3A) deSUMOylates FLS2 to negatively regulate immune signalling in the absence of flagellin. Yet, when flagellin is detected, Desi3A is degraded to enhance levels of SUMOylated FLS2 and increase immune signalling (Figure 1) [48]. In addition, it was found that monoubiquitination of BIK1 contributes to ligand-induced BIK1 dissociation from receptor FLS2 [123]. As mentioned previously, PUB12/13 triggers the degradation of FLS2 via the ubiquitin–proteasome system.

Post-translational modification on PTM machinery enzymes also occurs in defence; for example, CALCIUM-DEPENDENT PROTEIN KINASE 28 (CPK28) phosphorylases and activates Ub E3 ligases PUB25 and 26 to enhance ubiquitination and proteasomal degradation of nonactivated BIK1 (Figure 1) [89,124]. Interactions between Ub E3 ligases and the kinase domains appear to be common in the regulation of RLKs [125].

NONEXPRESSOR OF PATHOGENESIS-RELATED GENES (NPR1) is a key transcription factor in defence as it regulates the expression of *PR* genes contributing to the establishment of systemic acquired resistance (SAR) [126]. Again, phosphorylation, SUMOylation, and ubiquitination are essential for its function for appropriate defence responses (Figure 1). SUMOylation interacts with phosphorylation to control NPR1 functions: phosphorylation of Ser55 and Ser59 prevents NPR1 SUMO attachment. SUMOylation status of NPR1 alters its interaction with partners. Non-SUMOylated NPR1 interacts with WRKY70 to repress the expression of the *PR1*. On pathogen challenge, SA accumulation promotes dephosphorylation of Ser55/Ser59, allowing NPR1 to become SUMOylated, provoking NPR1 to interact with TGA3 to promote *PR1* gene expression [127,128]. Furthermore, NPR1 interaction with SUMO3 is required for Ser11/Ser15 phosphorylation, which causes ubiquitination and degradation by the NPR3–CULLIN3 E3 complex for specific and transient immune induction [129]. NPR1 degradation is important for the full range of defence gene activation and for activation of ETI and programmed cell death at the infection site, where SA levels are high [126], whereas in neighbouring cells SA levels are intermediate to allow NPR1 function [130]. SA-induced PR genes encode several antimicrobial metabolites including endoglucanases, chitinases, defensins, etc. [131]. This cited example displays that phosphorylation sites have opposing functions and that specific PTM patterns give outcomes in terms of defence response. The multiple-PTM sequential process provides more precise control to allow ubiquitin-mediated degradation at the right time when pathogens are not detected [132]. NPR1 has a functionally conserved role in crops; thus, potentially SUMO is involved in the regulation of orthologues similar to *Arabidopsis*, but this needs investigation [126,133].

Significant crosstalk exists between SUMOylation and ubiquitination, particularly as part of negative feedback to induce protein’s own degradation; for example, SIZ1 can SUMOylate CONSTITUTIVE PHOTOMORPHOGENIC 1 (COP1), which enhances the trans-ubiquitination activity of COP1, a multi-subunit E3 ligase which positively regulates disease resistance against viruses [134,135]. Following SUMOylation, COP1 ubiquitinates SIZ1 causes its degradation; therefore, ubiquitination regulates cellular SUMOylation by regulating SIZ1, as well as SIZ1 promoting COP1 ubiquitination activity [136].

Several SUMO targets overlap with MAPK phosphorylation targets in immunity regulation [137]. Several WRKYs were identified as targets of SUMO1 by proteomics, as well as MAPK phosphorylation [104]. To support this, it was exhibited that in response to *Botrytis cinerea* infection and flg22 elicitor treatment, WRKY33 is SUMOylated, which allows WRKY33 phosphorylation by MPK3/6 for activation of transcription factor activity leading to increased camalexin biosynthesis (Figure 1) [138].

It is clear that PTMs are vital for plant defence responses and disease resistance in *Arabidopsis*, and following this finding, PTMs are illustrated as similarly important in crop species and represent an excellent resource to be exploited in crop improvement. General mechanisms of immunity are similar in *Arabidopsis* and crop species along with classes of proteins; however, precise mechanisms, interactions, protein complexes, and PTMs are specific to the species and variety [17]. One study found that 1619 phosphosites in *Arabidopsis* aligned exactly to phosphosites of any other plant species, indicating some similarities in protein phosphorylation in *Arabidopsis* and crops [139]. In several cases, defence protein orthologues show a conserved role among different plant species; for example, PRRs, MAPK cascades, WRKY TFs, NPR1, ubiquitin ligases, and ubiquitination-mediated proteasomal degradation modulate defence protein accumulation [80,81,126,133,140,141,142,143,144].

In rice, differences in disease resistance may depend on the PTM pattern, as was suggested by the finding that the number and distribution of phosphorylation motifs differ between resistant and susceptible alleles of Pi54 [145,146]. Findings of PTM crosstalk in crops prove that PRR-mediated signalling in rice depends on specific phosphorylation patterns and ubiquitin-mediated control. The XA21 Thr705 residue is essential for rice PRR XA21 autophosphorylation. Thr705 is also essential for the interaction between XA21 and rice XA21 binding protein 3 (XB3) a ubiquitin ligase which is required for full XA21-mediated resistance [147,148]. This was demonstrated by the use of phospho-null mutant variants, XA21^T705A^ and XA21^T705E^, which are both unable to transduce the XA21-mediated immune response or interact with the XA21 binding proteins [147]. After PAMP perception by XA21 (which recognises *Xanthomonas oryzae* pv *oryzae* derived sulphonated peptides, [149]), XA21 specifically trans-phosphorylates XB3, which has been shown to auto-ubiquitinate in vitro, which may lead to activation of MAPK cascades [148,150]. The role of XB3 may be conserved between species in regulating cell death [151]. Additional to phosphorylation and ubiquitination, clearly specific SUMOylation regulation is essential in crops immunity since pathogen effectors pathogenicity by deSUMOylation [140]. The next section will describe in more detail how pathogens hijack the PTM systems for their own benefit, i.e., to evade host defences and gain nutrients to promote pathogen proliferation.

## 4. Effectors Disrupt Host PTMs

The regulation of PTMs is crucial for plant disease resistance to minimise pathogen establishment. Numerous pathogen effectors target PTM machinery, and some effectors themselves act as kinases, SUMO proteases, Ub E3 ligases, etc. to add/ remove PTMs, disrupting the plant defences for pathogen establishment (Table 1). Both aspects of disruption of the PTM-ome show that PTMs are one of the key components for modifying defence against pathogens.

Bacterial pathogens use a type III secretion system to inject effectors intracellularly. Several microbial effectors proteins act as E3 Ub ligases or interact with host E3 Ub ligases to disrupt host ubiquitination and regulation of targets [85]. One well-known case is the *Pst* type III bacterial effector AvrPtoB which has a C-terminal Ub ligase domain which ubiquitinates PRRs FLS2 and CERK1 causing proteasomal degradation, thereby suppressing defence [152,153]. AvrPtoB also causes proteasomal degradation of protein kinase Fen in susceptible tomato plants to prevent ETI activation [154]. In contrast, tomato Pto kinase, in the same family as Fen, interacts with AvrPtoB through binding of two domains of AvrPtoB to evade degradation and activate ETI in resistant tomatoes [155]. AvrPtoB also ubiquitinates and degrades NPR1 to disrupt SA defence signalling [173]. Clearly, pathogen effectors benefit by ubiquitin-mediated proteasomal degradation of immune components to suppress defence responses.

Effectors also target key immune signalling components. For example, a serine protease effector, HopB1, from *P. syringae* specifically cleaves the kinase-activated form of BAK1 [174]. Mutations of Arg297 and Gly298 inhibited the BAK1 kinase domain cleavage by HopB1, which explained why related protein SERK5 is not cleaved by HopB1 [174]. BAK1 is targeted by many effectors since it is a coreceptor to several PRRs [159,175]. Other effects act to disrupt host phosphorylation; for example, HopAO1 effector from *P. syringae* is a protein tyrosine phosphatase that dephosphorylates FLS2 and EFR to disrupt PTI [176]. HopAI1 effector inactivates MPK3, MPK4, and MPK6 through the removal of the phosphate group from phosphothreonine [163].

RPM1-INTERACTING PROTEIN 4 (RIN4) can regulate multiple immune signalling pathways and is targeted by four *P. syringae* effectors: AvrRPM1, AvrB, AvrRpt2, and HopF2 to disrupt RIN4 regulation [160,177,178]. Genetically, RIN4 acts as a negative regulator of PTI, but downstream of flg22 detection RIN4 is phosphorylated on S141 to derepress PTI [177]. One example where modification to PTMs leads to changes in disease resistance is with the phosphomimetic RIN4^T166D^ mutant which causes enhanced susceptibility to *PstDC3000*. RIN4^T166D^ plants exhibit inhibition of stomatal defences due to enhanced plasma membrane H(+)-ATPase activity allowing increased pathogen entry through stomata [177,179]. It was demonstrated that RIN4 phosphorylation at Thr-166 decreases as part of the defence response downstream of flagellin perception. However, *Pst* effector AvrB induces RIPK to phosphorylate RIN4 at Thr-166, which antagonises accumulation of the RIN4 S141 phosphorylated form, leading to PTI repression and *P. syringae* susceptibility (in susceptible genotypes which lack the relevant NLRs) [179,180]. RIN4 is guarded by NLR proteins in resistant genotypes, which recognise RIN4 modifications [178]. Pathogen effectors AvrRPM1 and AvrB induce RIN4 hyperphosphorylation of Thr-166 which reduces the RIN4-ROC1 interaction, which triggers the activation of NLR RPM1 [158,181,182]. AvrRpt2 proteolytically cleaves RIN4, and this is sensed by NLR RPS2 [183]. Activation of the RPM1 or RPS2 NLRs leads to ETI activation leading to HR and *Pst* resistance in plant genotypes containing these NLR genes [158]. The T166D RIN4 phosphomimetic is sufficient to induce RPM1 activation in resistant genotypes in the absence of pathogen effectors, showing the importance of this specific PTM [184]. Together, this shows that specific phosphorylation patterns are essential for RIN4 to act as a molecular switch to regulate two arms of defence [177]. The importance of PTMs and RIN4 is shown further since RIN4 is conserved in land plants, and S141 and T166 are evolutionarily conserved in RIN4 orthologues [184,185].

RIN4 is an intrinsically disordered protein, except in regions where pathogen-induced posttranslational modifications occur; the regions of disorder allow RIN4 to act as a signalling hub which can bind several different proteins which is important in signal transduction [179,185,186]. Substitution of a specific amino acid residue in RIN4 could potentially disrupt one or a few specific protein interactions to boost disease resistance. It was shown using circular dichroism spectroscopy that RIN4 phosphorylation affects protein flexibility; perhaps protein–protein interactions could be manipulated by using PTMs to influence RIN4 structure [179].

*Xanthomonas oryzae* pv. *Oryzae*, the causal agent of rice bacterial leaf blight produces the *Xanthomonas* outer protein K (XopK) effector which has E3 Ub ligase activity and directly ubiquitinates a PTI- related protein, rice somatic embryogenic receptor kinase 2 (OsSERK2), causing its degradation and disruption of PTI [165]. Mutation of the putative ubiquitin-conjugating enzyme (E2) binding site prevented XopK-induced degradation of OsSERK2 and disrupted XopK-dependent virulence [165]. *Xanthomonas euvesicatoria* (*Xe*) is the causal agent of bacterial spot disease of pepper and tomato and its effector XopAE also has E3 Ub ligase activity and inhibits plant immunity [187]. *Xe* type III effector XopAU acts as a protein kinase and disrupts host MAPK signalling through phosphorylation and activation of MKK2 [164].

*Xanthomonas* type III effector, XopD, has a C-terminal SUMO protease which removes SUMO from target proteins or processes SUMO precursors [140]. Tomato ethylene response factor (ERF) SIERF4 is targeted by XopD for deSUMOylation, causing SIERF4 destabilisation and ethylene production inhibition, which is required for ethylene-mediated immunity [166]. Surprisingly, XopD also can act as a SUMO and Ubiquitin isopeptidase [167].

XopD*_Xcc_*_8004_, a type III effector of *Xanthomonas campestris* pv. *campestris* (*Xcc*) 8004, a shorter form of the effector XopD which lacks the N-terminal domain, functions as a SUMO protease and this function is necessary to elicit host immune defences [188]. One target of XopD_*Xcc*8004_ deSUMOylation activity is HFR1, which is involved in the repression of plant defence responses.

Additionally, XopD*_Xcc_*_8004_ from *Xcc*8004 interferes with gibberellic acid (GA)-induced GA INSENSITIVE DWARF1 (GID1)-binding to hamper GA-GID1-DELLA complex formation and delay the induced ubiquitination and proteasomal degradation of DELLA protein, a repressor of ga1-3 (RGA). This influences the levels of DELLA proteins to minimise symptom development and promote disease tolerance [168]. XopD*_Xcc_*_8004_ is a suppressor of PTI through repression of the flg22-triggered ROS production [168]. The XopD*_Xcc_*_8004_DELLA interaction might be involved in this PTI suppression as DELLA is involved in SA and JA hormone defence responses [189]. Although XopD*_Xcc_*_8004_ contains the conserved putative cysteine protease SUMO domain of XopD effectors, deSUMOylation XopD*_Xcc_*_8004_ was not shown [168,169].

AvrBsT is a *Xanthomonas* YopJ-like effector, although YopJ-like effectors have homology with SUMO proteases, AvrBsT was identified to have acetyltransferase activity [170]. In pepper plants, AvrBsT targets proteasomal NON-ATPASE SUBUNIT 8 (RPN8) potentially to disrupt proteasomal function and targets energy sensor Sucrose nonfermenting 1 (Snf1)-related kinase (SnRK1) to disrupt the HR immune response elicited by effector AvrBs1 in resistant pepper plants [169,171]. AvrXv4 is another *Xanthomonas* YopJ-like effector which decreases the accumulation of SUMO-protein conjugates in *Nicotiana benthamiana* and pepper, *in planta* [172]. It may be that AvrXv4 has SUMO proteases activity, but it is not yet proven [169].

Fungal and oomycete effectors also act to affect the host’s ubiquitination system to evade immunity, for example, the *Phytophthora infestans* effector AVR3a, by modifying and stabilising host E3 Ub ligase CYS, MET, PRO, AND GLY PROTEIN 1 (CMPG1) to prevent the usual CMPG1 proteasomal degradation and prevent cell death [190]. The *Magnaporthe oryzae* (the causal agent of rice blast) fungal effector AvrPiz-t targets the RING E3 Ubiquitin Ligase AVRPIZ-T AND AVRPIZ-T INTERACTING PROTEIN 6 (APIP6) for degradation to suppress PTI in Rice [191].

The numerous effectors acting to disrupt immunity and promote pathogen establishment by altering plant PTMs demonstrate how crucial specific PTMs are to host resistance to pathogens. This ascertains that pathogen effectors disrupt PTMs in various ways, highlighting that precise regulation of PTMs is important in defence to prevent pathogen establishment.

## 5. Growth–Defence Trade-Offs

Plants tightly control the balance between growth and defence in order to optimise fitness and overcome stress [192,193]. Growth–defence trade-offs occur as plants restrict growth when activating their defence responses [194]; this could be to reallocate the plants’ limited resources when challenged by stress. Although in many cases, it is thought that resources are not a limiting factor, growth–defence trade-offs result from the careful regulation of complex signalling networks controlling plant metabolism [192,195,196,197,198].

SnRK1 and TOR (target of ramamycin) are energy sensors, act as global master regulators of metabolism, and play a dynamic and important role in the growth–defence balance [199,200]. SnRK1 and TOR are key in responding to biotic stress for plant survival (Figure 3) [201]. SnRK1 and TOR are protein kinase complexes which largely work antagonistically, and their crosstalk is evolutionarily conserved [198,202]. Typically TOR is activated in nutrient-rich conditions and promotes growth [203]. SnRK1, SnRK2, and SnRK3 subfamilies all have roles in promoting defence [204], but SnRK1 is the most prominent in the global regulation of metabolism in response to energy status [205]. SnRK1 is activated in response to energy depletion often occurring in stress conditions to restore energy homeostasis [206,207,208,209,210,211] (Figure 3).

PTMs are critical in the activities and regulation of SnRK-TOR growth–defence balance. SnRK1 and TOR phosphorylate targets to trigger transcriptional and metabolic reprogramming [212,213,214,215,216]. SnRK1 and SnRK2 repress TOR as part of their growth suppression by phosphorylating the regulatory-associated protein of TOR (RAPTOR) component; this regulation is evolutionarily conserved (Figure 3) [202,217]. SnRKs and TOR are integrated with hormone signalling, which can regulate growth [218]. For example, SnRK1 is a negative regulator of auxin-mediated primary root growth by activating *SHORT HYPOCOTYL 2/INDOLE ACETIC ACID 3 (SHY2/IAA3)* transcription [219], whereas auxin activates TOR signalling to promote growth [220].

SnRK1 is phosphorylated and activated by SnAK1 and SnAK2 (SnRK1-activating kinases), also known as geminivirus Rep-interacting kinases 1 and 2 (GRIK1 and GRIK2), which are regulated during plant development and geminivirus infection [221]. SnAK1 and SnAK2 have been shown to phosphorylate and activate the *Arabidopsis* SnRK1.1/SnRK1α1/KIN10 catalytic subunit on conserved residue Thr^175^. Phosphatases ABA INSENSITIVE 1 (ABI1) and TYPE 2A PROTEIN PHOSPHATASES (PP2CA) dephosphorylate and inactivate SnRK1 to regulate its activity [222].

SnRK1 provokes metabolic reprogramming under pathogen attack (Figure 3), which promotes broad disease resistance and plant fitness at the expense of growth, whilst TOR promotes growth and proliferation and suppresses defence-related genes, compromising immunity [198]. SnRK1 gain- and TOR loss-of-function plants tend to be more resistant, whereas TOR gain- and SnRK1 loss-of-function plants tend to be more susceptible; this is the case for viruses, bacteria, fungi, and oomycetes [198,223,224]. To support this, it was revealed that *OsSnRK1a* overexpression increased resistance against both (hemi)biotrophic and necrotrophic pathogens but suppressed normal growth and development, while *OsSnRK1a* silencing in RNAi lines increased susceptibility [225]. *OsSnRK1a* overexpression positively affected the SA pathway and boosted the JA defence to promote defence-related gene expression. TOR reduces plant defences by antagonising the action of SA and JA and suppresses defence-related genes [224,225]. SnRK1 is capable of phosphorylating viral proteins such as Rep to impair viral replication [226]. This highlights how significant SnRK1 is in defence responses. The regulation of SnRK1 and TOR can differ in different tissues [227,228].

SnRK1 is involved in enhancing immunity in a variety of ways through phosphorylation of targets. SnRK1 phosphorylates WRKY3, a repressor of immunity, to promote its proteasomal degradation, enhancing resistance to powdery mildew [229]. SnRK1 phosphorylation at Ser83 and Ser112 triggers WRKY3 degradation, and therefore, S83 and S112 mutated versions of WRKY3 were more stable than the wild-type protein. Homologue SnRK2.8 has a major role in regulating SAR, as its phosphorylation of monomeric NPR1 by SnRK2.8 at Ser-589 and possibly Thr-373 facilitates NPR1 entry into the nucleus [230]. Although SnRK2.8 activation is independent of SA, NPR1 monomerisation is triggered by SA-triggered redox changes [231,232]. SnRK1 is required for the induction of the AvrBs1-specific HR and programmed cell death (PCD) [171].

Several pathogens disrupt the SnRK1-TOR balance between growth and defence; for example, SnRK1 in rice is targeted by *Xanthamonas* effector AvrBsT (Table 1), showing that pathogens can disrupt this key plant defence regulator [171]. Likewise, viral suppressors of RNA-silencing proteins AL2 and L2 inhibit SnRK1 activity [208]. SnRK1 stability is also impacted by pathogens; the effector from *Fusarium graminearum*, the causal agent of Fusarium head blight, orphan secretory protein 24 (Osp24), accelerates the degradation of TaSnRK1α by facilitating its association with the ubiquitin-26S proteasome [233]. Similarly, the TOR pathway can be activated to benefit pathogens; for example, the *cauliflower mosaic virus* TAV effector protein binds to TOR, promoting its activity and leading to RIBOSOMAL PROTEIN S6 KINASE (S6K1) phosphorylation, which promotes translation reinitiation and viral replication [234]. However, favouring TOR pathway activity is not always beneficial to pathogen’s activity; the *Ralstonia solanacearum* effector AWR5 inhibits TOR signalling, perhaps to allow autophagy to proceed [235]. SnRK1 acts upstream of TOR as a positive regulator of autophagy in *Arabidopsis*, and TOR inhibits autophagy in nutrient-rich conditions through TOR-induced phosphorylation of AUTOPHAGY RELATED 1 and 3 (ATG1 and ATG13) proteins [236]. When TOR is inhibited, autophagy proceeds [237]. Selective autophagy cooperates with the ubiquitin–proteasome system to contribute to immunity but unregulated autophagy could benefit pathogens [238,239].

In addition to phosphorylation by SnRK1 and TOR kinases, regulation of growth-vs-defence also depends on SUMOylation, in conjunction with ubiquitination. The SnRK1 complex is SUMOylated at multiple positions by SIZ1 [240]. SUMOylated SnRK1 undergoes ubiquitination and proteasomal degradation to modulate SnRK1 signalling in *Arabidopsis*, whereas *siz1-2* null mutant and *siz1* catalytically inactive mutant show accumulation and hyperactivation of SnRK1. It was shown that SnRK1 triggers its own SUMOylation and ubiquitination-mediated degradation as part of a negative feedback loop; this ensures SnRK1 signalling is activated at the precise level, avoiding hyperactivation of defence responses. The dependence on SnRK1 activity controlling its own degradation was confirmed by the finding that phospho-inactive SnRK1α1 variants were not degraded as normal, but normal degradation of SnRK1α1 occurred in SnRK1α1 “SUMO mimetic mutants” mimicking the SUMOylated from of SnRK1 through translational fusion [240,241].

Downstream of SnRK1, via the domain of the unknown function (DUF)581-2, two DELLA proteins, gibberellic-acid insensitive (GAI) and RGA, were shown to be stabilised (Figure 4) [242]. DELLAs are suppressors of growth and act to suppress GA-responsive genes and GA biosynthetic genes and promote negative GA signalling components to maintain GA homeostasis [243]. *Arabidopsis* contains five DELLA protein genes (RGA, GAI, RGA-Like1 (RGL1), RGL2, and RGL3) which have some overlapping functions in repressing GA responses [242]. Stress signals inhibit the degradation by GA, including PAMP elicitor flg22 [244]. DELLAs are regulatory signalling hubs which integrate environmental signals and are regulated mostly at the post-translational level with SUMOylation, ubiquitination, and phosphorylation as critical PTMs of DELLAs (Figure 4) [245,246,247]. GA, a growth-promoting phytohormone, relieves the DELLA-mediated repression of genes through binding of GA to its receptor GID1 which triggers ubiquitination and proteasomal degradation of DELLAs [248,249,250,251,252,253,254].

SnRK1.1/SnRK1α1/KIN10 represses GA biosynthesis by phosphorylating and stabilising transcription factor FUS3 [255,256,257]. By contrast, TOR may promote GA signalling as mutants lacking in TOR component protein RAPTOR1B have decreased *GID1* expression and increased levels of DELLA protein RGA suggesting TOR may promote GA signalling [258]. Interestingly, around 28.6% of the genes induced by SnRK1.1 were also upregulated by DELLA protein RGA [259].

Stress signals, including pathogen infection, stabilise DELLA proteins preventing ubiquitin-mediated degradation which contributes to growth inhibition [189,244,245]. DELLAs cause susceptibility to biotrophs and resistance to necrotrophs by altering the balance of salicylic acid vs. jasmonic acid signalling in *Arabidopsis* [189]. In contrast to *Arabidopsis*, rice DELLA Slender Rice1 (SLR1) promotes resistance to (hemi)biotrophic but not necrotrophic rice pathogens [260]. Cassava (*Manihot esculenta*) MeDELLAs were shown as positive regulators of disease resistance against cassava bacterial blight [261]. This shows that DELLAs are important positive regulators of defence in diverse species.

SUMOylation of DELLA occurs in stress, and the SUMOylated DELLA binds to GID1 via its SUMO interacting motif (SIM). This occurs independently of GA, which sequesters GID1 to prevent GA degradation (Figure 4) [262,263]. This leads to an accumulation of non-SUMOylated DELLA which causes repression of GA responses and growth restriction. Some phenotypes of *Arabidopsis* SUMO protease mutant *ots1ots2* are mediated through DELLA since the knockout of a DELLA protein restores the *ots1ots2* double mutant background to the WT phenotype [264]. Higher DELLA levels accumulate in the *ots1ots2* double mutant, which shows that OTS1/2 deSUMOylate DELLA, which destabilises DELLAs. However, DELLA stabilisation causes high DELLA levels and reduced fertility [264]. Rice DELLA SLR1 also undergoes SUMOylation, which alters its interaction with specific transcription factors to improve abiotic stress tolerance [265]. There is a suggestion that SLR1 SUMOylation may attenuate the penalty of salt stress tolerance on plant yield [265], with the goal of maintaining yield and disease resistance under pathogen stress in rice, which would be interesting to explore. In addition, mutation of the SIM site in GID1 in rice or *Arabidopsis* could be manipulated to fine-tune DELLA degradation [263]. DELLA stability is also increased by phosphorylation (Figure 4); in rice, EARLIER FLOWERING 1 (EL1) stabilised SLR1 [266]. In *Arabidopsis,* it was shown that protein phosphatase dephosphorylates DELLA promotes GA-induced degradation [267].

DELLA protein RGL3 positively regulates JA-mediated resistance to the necrotrophs [268]. DELLAs promote JA defence responses by competing with MYC2 for binding to JAZ proteins; this relieves MYC2 from JAZ suppression to allow MYC2-dependent JA responses to contribute to the balance of growth and defence [269,270]. Similarly to protein JA responses, MdSnRK1.1 phosphorylates MdJAZ18 protein in apple to facilitate its 26S proteasome-mediated degradation which is likely relevant in defence [271]. Intriguingly, SnRK1 mediates proteasomal binding of a plant SCF ubiquitin ligase which can modulate JA responses. Pathogen infection stabilises DELLA proteins RGA and RGL3 to restrict growth in a partially EDS1-dependent manner [244]. However, DELLA also directly interacts with EDS1 to together decrease SA production as part of a negative feedback mechanism to modulate the SA accumulation and to prevent excessive defence response (Figure 4) [244]. Clearly, DELLAs alter the balance of salicylic acid vs. jasmonic acid signalling, and DELLA regulation by PTMs is important in growth–defence balance [189].

There may be potential to control specific elements of this growth-defence network through manipulation of SnRKs/TOR phosphorylation targets or through other interacting PTMs, to uncouple antagonistic activities in growth and defence to yield [272]. Beyond SnRK1 vs. TOR antagonism, several other components have antagonistic pathways to balance growth and defence; for example, the MAPK cascade MEKK1-MKK1/2-MPK4 negatively regulates plant cell death and immunity downstream of PAMP activation of PRRs, whilst MPK3/6 cascades positively regulate immunity [61].

Plants mitigate growth–defence trade-offs through methods including inducible tissue-specific defence and priming [273]. Defence pathways can be “primed” for faster and stronger activation to subsequent pathogen attacks, and primed states can be transmitted to offspring [274]. Priming by elicitors such as flg22 and chitin could be mediated by manipulating PTMs on components such as NPR1. MAPKs could potentially induce a primed state, but more investigation is necessary [275]. Changes in phosphorylation could potentially change the growth–defence balance after priming. Using priming agent Β-AMINOBUTYRIC ACID (BABA), a mutation in eIF2α-phosphorylating GENERAL CONTROL NON-DEREPRESSIBLE 2 (GCN2, also known as PBL27) kinase did not affect BABA-induced immunity, but relieved BABA-induced growth repression [276]. Interestingly, TOR blocks the action of GCN2 to promote translation, since GCN2 inhibits translation initiation upon sensing the uncharged transfer RNAs that accumulate during amino acid limitation to maintain amino acid homeostasis in nitrogen deficiency [277]. GCN2 function might be conserved between plant species [206].

Hormone metabolism must be tightly controlled in the right situation, with most plant hormones being involved and interacting in immunity [278]. SA and JA are typically antagonistic, although SA and JA can occasionally act synergistically as well [279]. Several hormones control the balance between growth and defence: auxin and SA are antagonistic, with auxin promoting growth, and SA promoting defence [280,281]. JA inhibits growth as part of defence [282], and crosstalk exists between brassinosteroid, auxin, and gibberellin signalling.

Inhibition of photosynthesis is frequently observed as part of the defence response; reducing photosynthesis may starve biotrophic pathogens of nutrients [283]. However, mutant *jazQ (jaz quintuple) phytochromeB (phyB)* plants grow and defend well simultaneously; the whole-plant photosynthetic rate in *jazQ phyB* plants was similar to WT, showing that perhaps manipulation of plant proteins can alter the balance of growth and defence and that hormone pathways are important.

BR-mediated growth can antagonise innate immune signalling [284]. Yet, treatment with Brassinolide (BL), the main brassinosteroid, induced resistance to a range of diseases in tobacco, and resistance to rice blast and bacterial blight in rice [285]. Additionally, BRs can increase resistance to the cucumber mosaic virus [286]. BR treatment increases resistance to necrotrophs and insects via increased JA response [287]. BAK1 is involved in PTI and in brassinosteroid signalling in development, whereas BIK1 positively regulates plant immunity, yet negatively regulates BR signalling [288]. As mentioned in a previous section, BAK1 mutants T450A and C408Y both show severe defects in immune defence but normal growth phenotype, proving that the phosphorylation patterns of RLK partners by BAK1 could selectively regulate multiple BAK1-dependent pathways [46]. Interestingly, the gain-of-function bak1^elg(elongated)^ protein results in increased BR signalling and impaired response to flagellin [289]. Further gain-of-function mutations could potentially increase defence signalling without affecting growth, although BRs antagonise immunity without BAK1 [284].

Significantly, *IDEAL PLANT ARCHITECTURE1 (IPA1)*/*WEALTHY FARMER’S PANICLE* (*WFP)*/*Rice SQUAMOSA PROMOTER BINDING PROTEIN-LIKE 14 (OsSPL14*) was identified to enhance yield-related growth as well as disease resistance, and PTMs are crucial to its regulation [290,291]. *OsSPL14* positively regulates panicle branching and grain numbers per panicle in the reproductive stage and negatively controls shoot branching (tillering in rice) in the vegetative stage, by regulating the expression of *TEOSINTE BRANCHED1 (TB1)* and *DENSE PANICLE 1 (DEP1)* [290,292]. Phosphorylation and ubiquitination are necessary for OsSPL14 activity and regulation. OsSPL14 is phosphorylated at the serine163 residue following pathogen infection, which changes its DNA binding specificity to activate the expression of *WRKY45*, which then enhances disease resistance [291]. OsSPL14 returns to the nonphosphorylated state within 48 h postinfection to activate genes related to growth and high yield [291]. A RING-finger E3 Ub ligase, IPA1 INTERACTING PROTEIN1 (IPI1), carries out tissue-specific ubiquitination which promotes the degradation of OsSPL14 in panicles, whilst stabilising OsSPL14 in shoot apexes [293]. This is caused by IPI1 ubiquitinating OsSPL14 with different polyubiquitin chains, adding K48-linked polyubiquitin chains in panicles for OsSPL14 degradation, and K63-linked polyubiquitin chains in the shoot apex to control plant architecture [293]. The natural *ipa1-1D* allele has a nucleotide substitution at the OsmiR156 target site, allowing it to resist microRNA transcript cleavage, resulting in higher expression in panicles [294,295]. This allowed a 10% yield increase without blast disease, up to 40% with blast disease than controls in field trials [291]. Overexpression of *IPA1/OsSPL14* also enhanced disease resistance against bacterial blight but a reduction in yield was observed; however, the yield was restored when expressing *OsSPL14* with the pathogen-inducible promoter of OsHEN1 [296]. This phosphorylation switch to defence gene expression which is reversed after 48 h is essential for OsSPL14 function, combined with tissue-specific ubiquitination controlling stability. This K48 vs. K63 ubiquitin linkage needs more investigation to investigate how widespread this regulation is, along with its potential for manipulation.

SPL protein homologues have different functions but share a highly conserved DNA-binding domain (SQUAMOSA-PROMOTER BINDING PROTEIN (SBP) domain) and a conserved serine residue which functions as a phosphorylation site [297]. Phylogenetic analysis identified that SPL subgroup III contained orthologous SPL proteins, including OsSPL14 (IPA1), OsSPL7, and OsSPL17 from rice; ZmSBP8, ZmSBP30, and ZmSBP6 from maize, and AtSPL9 and AtSPL15 from *Arabidopsis*, which all perform a similar function in regulating vegetative/reproductive branching in various plant species [298,299,300]. In *AtSPL9*-overexpressing plants, there was a greater accumulation of ROS and transcripts of basal salicylic acid signalling pathway genes, compared with wild-type Col-0 plants; thus, AtSPL9 could have a role in disease resistance [301]. These maize SPL proteins could be investigated to discover if the phosphorylation–ubiquitination control has any roles in disease resistance and growth, similar to OsSPL14 (Appendix A).

To improve plant disease resistance, optimising the balance between growth and defence is important. Growth trade-offs may not be inevitable with increased immunity and disease resistance with the right strategy [195,197]. It is critical to improve disease resistance to reduce pathogen colonisation and crop losses whilst minimising compromises in growth and reproduction to maintain and maximise yield in a dynamic environment. Alteration of specific protein PTMs could potentially promote certain interactions to allow enhanced disease resistance and growth simultaneously whilst allowing growth at the right times; methods will be explored in the next section.

## 6. Exploiting PTMs to Produce Disease-Resistant Crops

Crop selection is based on yield-related traits; the diversity of disease resistance genes in most crop plants today has been reduced as a consequence [302]. NLR genes have been used widely but are often not durable as a result of pathogen evolution. The pyramiding of NLR genes can be a solution to durability but can cause reduced growth and yield in the absence of pathogen infection [296]. NLR gene introduction can lead to excessive HR response, inappropriate activation of defence genes, or regulation of ROS [65,303,304,305]. There needs to be more research into durable disease resistance without compromising yield. Several potential advances need testing in the future. Genome editing, specifically the clustered regularly interspaced short palindromic repeats/CRISPR-associated protein (CRISPR/Cas) system, has the ability to generate knockouts of, for example, “susceptibility genes”; however, this can have detrimental effects if a protein is multifunctional [306]. The use of CRISPR/Cas and knowledge of PTMs enable the change in critical effector interaction residues of pathogen targets to prevent pathogen PTM attachment and methods of pathogenicity (Figure 5).

Manipulation of PTMs can be exploited to increase plant disease resistance; for example, in rice, overexpression of the phosphomimetic version of OsWRKY53 enhanced resistance to rice blast, compared to overexpression of the WT version of OsWRKY53 [307]. The MAPK cascade OsMKK4-OsMPK3/OsMPK6, which functions in the response to fungal PAMPs in rice, phosphorylates the SP (serine-proline) cluster of OsWRKY53 in vitro and likely in vivo [307]. The SP cluster is a highly conserved cluster among several group-I WRKY proteins in higher plants, and the phosphomimetic version of OsWRKY53 has all six Ser residues in the SP cluster substituted for Asp (OsWRKY53SD) which mimics phosphoserine [308]. Coexpression of OsWRKY53 with a constitutively active OsMKK4 increased OsWRKY53 transactivation activity in an SP cluster-dependent manner; furthermore, the OsWRKY53 phosphomimetic had enhanced transactivation activity compared to the WT version. These together suggest that phosphorylation of the SP cluster increases transactivation activity [307]. Interestingly, phosphorylation of OsWRKY53 by OsMPK6 did not alter its DNA-binding activity to W-box elements. The phosphomimetic *OsWRKY53SD-OX* rice plants had enhanced defence to rice blast and high activation of defence genes, including PR genes, compared to *OsWRKY53-OX* plants. Additionally, it was found that plants overexpressing the phosphomimetic OsWRKY53 have normal growth and development. This strategy has potential for crop production; however, yield tests, followed by large-scale field trials, would need to be carried out. PTM mimics are not perfect and therefore could have unexpected effects [307].

PTMs that a protein undergoes depends on protein sequence as well as other modes of regulation. Therefore, genetic sequence variants influence the protein sequence and hence PTMs. Sequence variants can be associated with disease resistance through PTMs. SNP databases are starting to increase particularly for rice, and there are trait-associated SNP databases for stress-related SNPs [309,310]. Prediction of how SNPs may influence PTMs is improving, which could be useful to predict protein interactions, enzyme activity, and protein turnover of different gene alleles, as well as to direct hypotheses, find out the specific mechanisms, and understand potential pleiotropic effects [311]. In animals, disease-associated PTM-SNPs have been identified and assembled into a database; thus, a similar idea could be formed for crop plants [312,313].

Biotechnological advances allow a proteome-wide approach for discovery, as well as a rational, targeted approach in the production and testing of new crop lines. Biotechnological approaches also extend beyond the reliance on existing natural allelic variations present in sexually compatible germplasm [314]: the application of genetic engineering is one of the leading technological advances in recent decades [315]. Genetic engineering has typically involved knockout or overexpression of genes to modify defence response pathways, but this can cause yield and/or quality trade-offs. There is the challenge to avoid these growth and yield trade-offs with improvements in disease resistance [316]. Finally, genome editing, in particular CRISPR/Cas, has become the most important biotechnological tool which had great potential and is recently being increasingly utilised [317].

Base editors or prime editors, as part of the CRISPR/Cas system, will be useful to modify critical PTMs of defence-related proteins through modification of nucleotides to alter specific amino acids. Alteration of amino acids could increase the stability of PTMs or abolish PTMs, thus altering protein function, interaction, and downstream responses to produce disease-resistant plants and crops (Figure 5). For example, alteration of ubiquitin site could increase the stability of immune signalling components, so long as they are inactive until activated by subsequent phosphorylation/SUMOylation at the appropriate time to avoid growth penalties.

Base editors could prove useful for making amino acid substitutions in defence-related proteins. Base editing does not involve double-strand breaks and features a Cas9 nickase (Cas9n) (or catalytically inactive Cas9) fused to nucleoside deaminases [318,319,320]. Cytosine base editors (CBEs) and adenine base editors (ABEs) currently enable four types of nucleotide conversions (C to T, T to C, A to G, and G to A) [321]. Recently using engineered Cas9 variant, Cas9-NG, fused to base editors, rice *BR-SIGNALING KINASE 1 (OsBZR1)* gain-of-function mutants carrying C > T conversions were successfully identified. Additionally, A > G conversions were induced in *OsSERK2* with a 9–40% success rate [321]. The A > G conversion targeted phosphorylation site in OsSERK2, which is expected to change downstream signalling in development or defence [322]. 

The C > T conversions carried out in OsBZR1 caused a P234L substitution, which is predicted to produce the ortholog of the stabilising gain-of-function *Arabidopsis* allele, *bzr-1d*. Plants with the *bzr-1d* allele have increased BZR1 dephosphorylation, enhanced BR signalling, and BR-mediated growth [323,324]. This BZR1-1D mutation has been reported to increase tomato quality [325] and the *bil1-1D/bzr1-1D* allele increased resistance to thrips in *Lotus japonicus*, which cause damage and transmit disease possibly through increased JA levels [287]. BZR-1D allele could potentially balance growth and yield since increased BR-signalling leads to increased seed production [326]. However, disease resistance must be maintained despite the increased BR signalling, potentially by using combinations of promoters, coding sequences, vectors, and genotypic backgrounds, which is complex and thus needs more research [324].

Editing OsSERK2 is promising for balancing defence and yield, as OsSERK2 regulates brassinosteroid-mediated growth and PRR immune signalling [322]. Specific phosphorylation sites in AtBAK1 mediate interactions and responses; thus, it is likely this is the case in crops [46]. OsSERK2 positively regulates immunity mediated by XA21, XA3, and OsFLS2 which are structurally similar receptor kinases [322]. *OsSERK2* is required for rice *Xa21*-mediated resistance to *Xanthomonas oryzae* pv. *oryzae* (*Xoo*) and to the hemi-necrotrophic fungus *Magnaporthe oryzae*. OsSERK1 (OsBAK1) has greater similarity to AtBAK1 and is important in plant growth and development, but OsSERK1 is not required for rice immunity to *Xoo* or *M. oryzae* [327]. OsSERK2 undergoes bidirectional transphosphorylation with XA21 in vitro and forms a constitutive complex with XA21 in vivo, unlike BAK1’s interaction with FLS2 and EFR which occurs after ligand binding, and BAK1 carries out transphosphorylation rather than FLS2 or EFR [322]. The phosphorylation pattern effects must be explored further.

Prime editing is an exciting new tool which allows the introduction of all mutation types, including insertions, deletions, and all putative 12 types of base-to-base conversions [328]. Prime editors, which are CRISPR–Cas9 nickase-reverse transcriptase fusions programmed with prime editing guide RNAs (pegRNAs), can edit bases without donor DNA or double-strand breaks and have been demonstrated in rice and wheat cells [328,329]. Mutations identified or predicted to improve disease resistance can be achieved in crops with prime editing techniques, although efficiency needs improvements. Ser/Thr phosphorylation sites, Ubi, and SUMO lysines are replaced by alanine to disrupt the site, but also point mutations elsewhere in the protein can alter protein structure or interactions and therefore function [46,118,291]. Substitution of the important residues surrounding a PTM site, rather than the attachment amino acid site, could weaken or strengthen PTM attachment, rather than completely abolishing it. The substitutions could alter the strength of the enzyme–substrate interactions or other signalling interactions [56,146,330].

One strategy for improved disease resistance could be to prevent effector post-translational modification; genome editing could be utilised to modify critical amino acid residues targeted or induced by pathogen effectors to undergo PTM attachment (Figure 5) [185,306]. For example, base editing or prime editing could be used to substitute critical PTM sites in RIN4 to boost immunity in crop plants. For instance, one way could be to modify the conserved Thr-166 phosphorylation site to overcome the susceptibility to pathogen effectors including AvrB and Rpst2. Thr-166 counteracts the flg22 induced phosphorylation of S141 to suppress defence when it is activated: the T166A phospho-null RIN4 mutant still maintained the flg22-activated suppression of *Pseudomonas syringae pv. tomato strain DC3000* proliferation [177]. Similarly, T166A substitution may not lead to overactivation of defence since RIN4 is a negative regulator of PTI until S141 phosphorylation causes derepression of PTI [177,185]. 

Pathogen-inducible promoters such as *OsHEN1* [296], *OsCYP76M7* [331], and *TBF1 (TL1-BINDING FACTOR)* [332] may prove useful to overexpress PTM machinery enzymes including kinase/phosphatases and SUMO proteases that are positive regulators of immunity, specifically under stress, or in specific tissues to boost disease resistance at the time needed to avoid growth cost associated with constitutively activated defence responses, for example, to enhance SnRK1 expression specifically in stress affected regions or reduce expression of PUB12/13 specifically in stress to relieve the autoimmunity of the *pub13* mutant [86]. CRISPR/Cas techniques can be utilised for sequence replacements using HOMOLOGY DIRECTED REPAIR (HDR) to replace or edit a specific promoter for stress-inducible or tissue-specific genes [333,334]. However HDR sequence replacement has low efficiency in plants; therefore, efficiency needs to be improved.

Catalytically inactive or dead-Cas9 (d-Cas9) driven by an inducible or tissue-specific promoter could prevent transcription of particular genes of PTM machinery to boost immunity with pathogens detected in a particular cell type. This could reduce transcription of a PUB ubiquitin ligase, SUMO protease, specific kinase/phosphatase to control specific PTM attachment/removal to reduce specific defence protein activation/inactivation or turnover during the pathogen stress [335]. SUMO proteases and ubiquitin ligases provide specificity in their respective pathways, and therefore, modification of these enzymes expression could provide a more specific response [48,336,337,338].

The use of genome editing speeds up crop breeding for specific sequences that generate beneficial traits. Gene editing can also introduce new edits not found within reproductively compatible germplasm, but unlike genetic engineering, the transgene for the CRISPR construct can be segregated out after stable integration or can be delivered transiently [339].

## 7. Conclusions and Perspectives

All aspects of plant defence use PTMs; the significance of PTMs is also clear since pathogen effectors disrupt PTM regulation as part of their virulence. Strategies predicted to increase disease resistance include the substitution of specific amino acid residues to stabilise or destabilise the formation of specific PTMs for precise control of protein function (Figure 5). One challenge will be to improve the efficiency of CRISPR/Cas techniques such as base editors and prime editors which, at present, show low efficiency. The growth–defence balance is critical—favouring defence does not necessarily mean that growth must be penalised if a careful manipulation of PTMs to control protein interactions is considered. One major challenge is translating lab research into field crops which can experience diverse and changeable conditions. Field trials must be conducted, as increased resistance to a particular disease could possibly have negative effects on the plants’ ability to respond to different types of pathogens, abiotic stress, beneficial microorganisms or could impact crop quality [114,340,341,342,343]. Yet, importantly, knowing the detailed mechanisms of disease resistance should increase the success of new resistant crop lines in the field.

It is still technically difficult to identify and prove protein–PTM functions in vivo, but advances in the sensitivity of proteomics will improve the detection of the numerous PTMs that are dynamic or occur in low stoichiometry [42,54]. Protein phosphorylation is the best-characterised PTM in plants thus far, with several databases available, including the PTM viewer which identified 326,848 sites in 89,022 proteins [139,344]. One limitation is that SUMO and ubiquitin attachment site prediction can be difficult since not all SUMO sites match the consensus motif [345], and the pattern of ubiquitination sites is not conserved in different species [346]. Intriguingly, ubiquitin itself undergoes phosphorylation and other PTMs; this, combined with the architecture of ubiquitin chains, makes ubiquitination more complex [78,347]. Ubiquitin and SUMO attach to lysine residues, but other PTMs also attach to lysine, such as lysine acetylation with histone and nonhistone protein acetylation functions emerging [348,349]. Acetylation regulation is important in defence; for example, fungal and bacterial effectors disrupt host acetylation to promote virulence, such as AvrBsT which acetylates proteins such as ACIP1 to alter their defence function [170,350] (Table 1). Lysine acetylation is reversible, in contrast to N-terminal acetylation which regulates NLR protein SNC1 stability, possibly via the ubiquitin-mediated proteasomal system [117]. Thus, PTM crosstalk in plant immunity needs further exploration.

Notably, both protein–protein interactions and PTMs often depend on small regions of one or a few crucial amino acid residues [351]. In the future, CRISPR/Cas base editors or prime editors [318,329] have great potential to produce precise and targeted point mutations, to change single amino acids, to alter a specific interaction of a multifunctional protein to enhance disease resistance whilst avoiding negative effects, to reduce the serious problem of crop losses from disease.

Figure 1, Figure 2, Figure 3, Figure 4 and Figure 5 were created with Biorender.com.

## Figures and Tables

**Figure 1 biomolecules-11-01122-f001:**
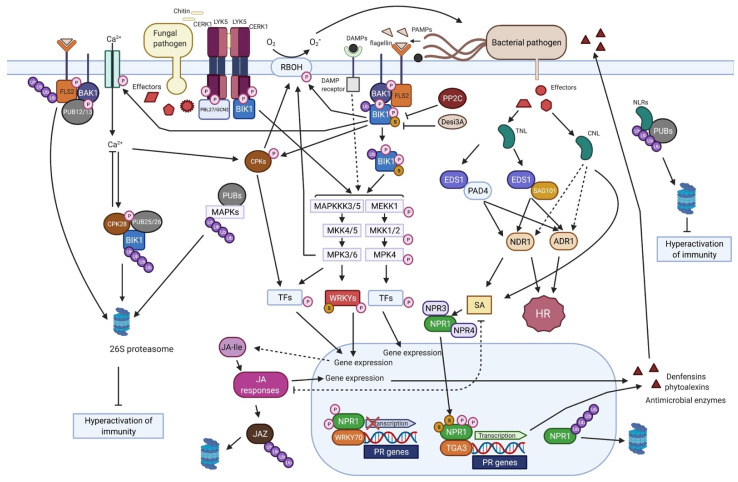
The framework of plant immunity. Briefly, microbe-derived pathogen-associated molecular patterns (PAMPs) and host damaged-associated molecular patterns (DAMPs) are perceived by cell-surface pattern recognition receptors (PRRs), which along with coreceptors activate downstream phosphorylation cascades and induce increased [Ca^2+^] and reactive oxygen species (ROS) accumulation. ROS can act as second messengers to cause stomatal closure and directly have antimicrobial effects. Pathogen effectors are perceived by intracellular receptors, NUCLEOTIDE BINDING SITE-LEUCINE-RICH-REPEAT (NLRs also known as NB-LRRs), which activate downstream responses including Salicylic acid (SA)accumulation. SA and Jasmonic Acid (JA) accumulate in response to pathogens and largely work antagonistically to biotrophic and necrotrophic pathogens, respectively. Outcomes of defence signalling include changes to gene expression, production of (PR) proteins, and biosynthesis of antimicrobial metabolites. Post-translational modifications (PTMs) are involved in all aspects of plant immunity controlling activation, protein interactions, subcellular localisation, and protein turnover. Further details in the main text. P, phosphate group. S, SUMO. Ub, Ubiquitin. Solid lines indicate direct interactions, dashed lines indicate indirection interactions. FLS2, FLAGELLIN-SENSING 2; BAK1, BRI1-ASSOCIATED RECEPTOR KINASE; BIK1, BOTRYTIS-INDUCED KINASE 1; EDS1, ENHANCED DISEASE SUSCEPTIBILITY 1; CERK1, CHITIN ELICITOR RECEPTOR KINASE 1; LYK5, LYSM-CONTAINING RECEPTOR-LIKE KINASE 5; NPR, NONEXPRESSOR OF PATHOGENESIS-RELATED GENES; TGA3, TGA 1a-related gene; CPK, CALCIUM-DEPENDENT PROTEIN KINASES; NDR1, NON-RACE-SPECIFIC DISEASE RESISTANCE 1; ADR1, ACTIVATED DISEASE RESISTANCE 1; PAD4, PHYTOALEXIN DEFICIENT 4; SAG101, SENESCENCE-ASSOCIATED GENE 101; PUB, PLANT U-BOX; HR, hypersensitive response; JAZ, JASMONATE-ZIM-DOMAIN PROTEIN 1; Desi3A, DeSUMOylating isopeptidase 3A, PPCA, PROTEIN PHOSPHATASE TYPE 2C. PBL27/GCN2, GENERAL CONTROL NONREPRESSED 2, RBOH, respiratory burst oxidase homolog; TFs, transcription factors; JA-Ile, jasmonoyl–isoleucine; TNL, Toll-interleukin-1 receptor-like nucleotide-binding site leucine-rich repeat; CNL, coiled-coil (CC)-NBS-LRR. Figure created with Biorender.com.

**Figure 2 biomolecules-11-01122-f002:**
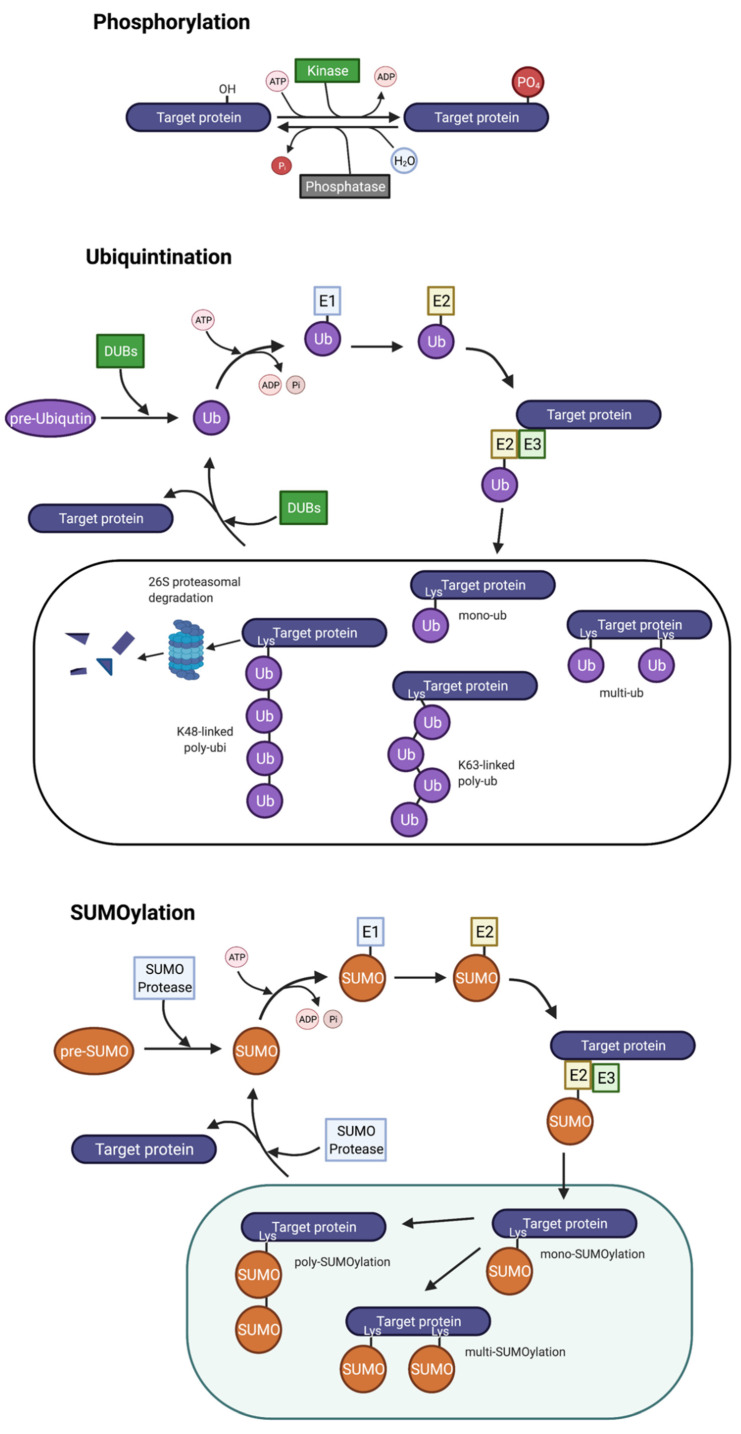
Post-translational modification pathways. Phosphorylation is the process catalysed by protein kinases in which a phosphate group (PO_4_) is transferred from ATP onto the side-chain hydroxyl groups on serine, threonine, or tyrosine residues on a target protein. Phosphatases hydrolyse the phosphoester bond to remove the phosphate group. Ubiquitination involved the sequential action of the ubiquitin-activating enzymes (E1), ubiquitin-conjugating enzymes (E2), and ubiquitin-protein ligases (E3) to covalently attach ubiquitin onto the target lysine. Different ubiquitin attachment linkages and chain lengths have different functions; for example, K48-linked tetraubiquitin targets the protein for 26S proteasomal degradation. Deubiquitinating enzymes (DUBs) catalyse deubiquitination. SUMOylation is analogous to ubiquitination and involves the sequential action of SUMO E1, E2, E3 enzymes to covalently attach SUMO onto the target lysine. SUMO is synthesised as an inactive precursor which has its C-terminal peptide cleaved by a SUMO protease exposing the di-glycine motif. SUMO proteases also catalyse the removal of SUMO.

**Figure 3 biomolecules-11-01122-f003:**
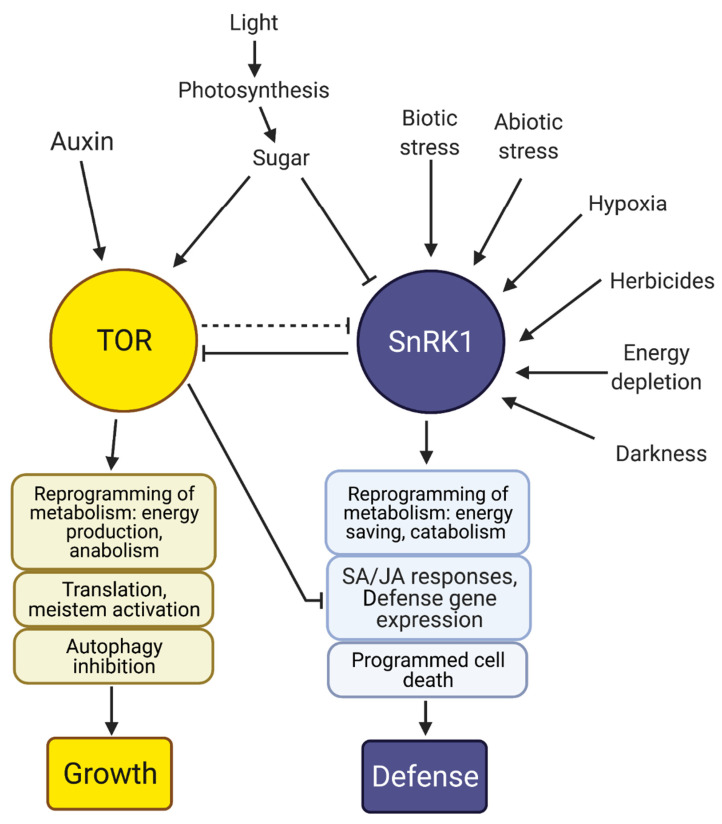
A simplified model of SnRK1-TOR growth–defence regulation. Kinases Sucrose nonfermenting 1 (Snf1)-related kinase (SnRK1) and target of ramamycin (TOR) are master regulators which sense energy status and work antagonistically to reprogramme metabolism through phosphorylation of diverse targets. TOR is active in nutrient-rich conditions to promote translation and growth while inhibiting autophagy. SnRK1 is activated in times of energy depletion often caused by stress, operates to promote defence responses, and suppresses growth. SnRK1 phosphorylates and inactivates TOR directly to limit growth and promote autophagy. TOR inhibits SnRK1 outputs indirectly. Solid lines indicate direct interactions, dashed lines indicate indirect interactions.

**Figure 4 biomolecules-11-01122-f004:**
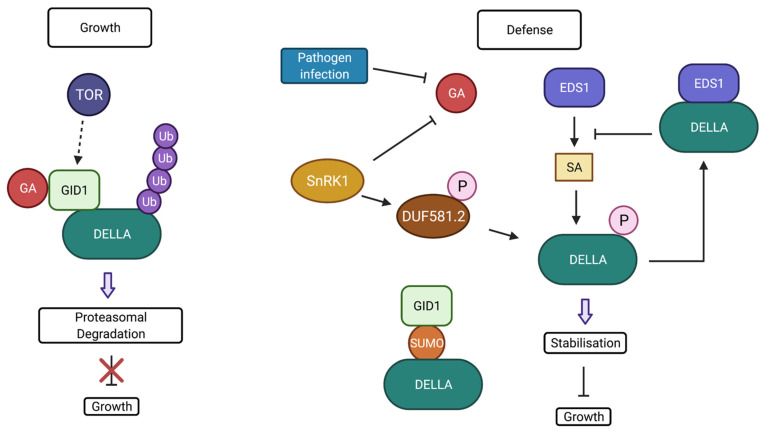
DELLA interactions with PTMs in growth and defence. When gibberellic acid (GA) accumulates, it binds to GA INSENSITIVE DWARF1 (GID1) which then binds to DELLA, triggering DELLA ubiquitination and proteasomal degradation and allowing GA-responsive gene expression and growth. DELLA is stabilised in various ways to promote defence and restrict growth. In defence, GA levels are reduced which reduces GA-mediated DELLA degradation. DELLA is stabilised by phosphorylation. Independent of GA, DELLA is stabilised by SUMOylation which blocks GID1 degradation on unSUMOylated DELLA. SnRK stabilises DELLA through intermediate protein DUF581-2. DELLA forms a negative feedback loop to control SA accumulation. P, phosphate group. S, SUMO. Ub, Ubiquitin. Solid lines indicate direct interactions; dashed line indicates suggested positive interaction.

**Figure 5 biomolecules-11-01122-f005:**
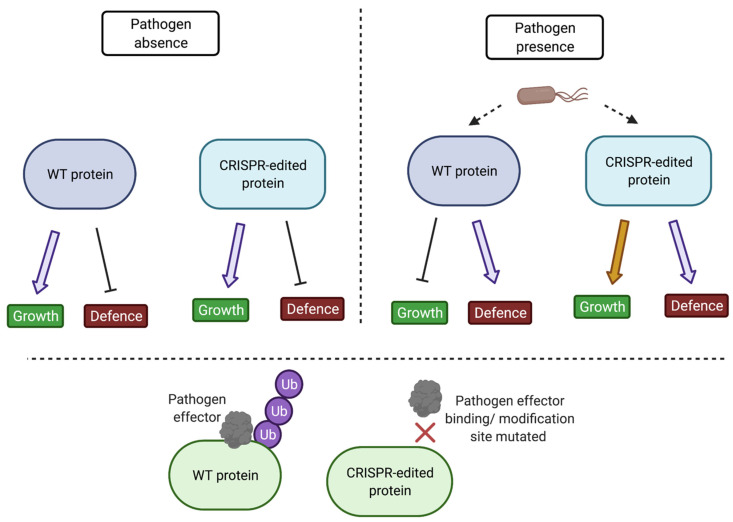
Model for improving disease resistance centring on PTMs. Genome-edited proteins giving rise to amino acid substitutions may confer disease resistance by allowing the protein to evade pathogen effectors, or through optimising growth–defence trade-offs to avoid excessive growth restriction in defence by controlling precise protein–protein and protein–PTM interactions.

**Table 1 biomolecules-11-01122-t001:** Effector examples which regulate host PTM systems.

Effector	Pathogen	Target/Host	Function	References
**AvrPtoB**	*P. syringae*	FLS2, BAK1, CERK1 (Arabidopsis), FEN (Tomato)	E3 ubiquitin ligase	[152,153,154,155]
**AvrPto**	*P. syringae*	FLS2, EFR (Arabidopsis) LeFLS2, RIN4 (tomato)	Kinase inhibitor	[156,157]
**AvrRpm1**	*P. syringae*	RIN4 (Arabidopsis)	Induces phosphorylation	[158]
**AvrB**	*P. syringae*	RIN4 (Arabidopsis)	Induces phosphorylation	[158]
**HopF2**	*P. syringae*	RIN4, BAK1, MKK5 (Arabidopsis), MPK6 (Tomato)	ADP-ribosylation	[159,160,161,162]
**HopAI1**	*P. syringae*	MPK4, MPK6 (Arabidopsis and tomato)	Phosphotheonine lyase	[161,163]
**XopAU**	*Xanthomonas* sp.	MKK2 (tomato)	Protein kinase	[164]
**XopK**	*Xanthomonas* sp.	OsSERK2 (rice)	E3 ubiquitin ligase	[165]
**XopD_*Xcv*85-10_**	*Xanthomonas* sp.	ERF4 (tomato)	SUMO protease, ubiquitin protease	[140,166,167]
**XopD_*Xcc*8004_**	*Xanthomonas* sp.	RGA (Arabidopsis)	SUMO protease?	[168]
**AvrBsT**	*Xanthomonas* sp.	SnRK1, ACETYLATED INTERACTING PROTEIN1 (ACIP1), proteasomal subunit RPN8 (pepper)	Acetyltransferase	[169,170,171]
**AvrXv4**	*Xanthomonas* sp.	Unknown targets	SUMO protease?	[169,172]

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
