# Peer review of "Understanding and Exploiting Post-Translational Modifications for Plant Disease Resistance"

_biomolecules, 2021, doi:10.3390/biom11081122_

Round 1

Reviewer 1 Report

The topic of this review is important and timely. Authors nicely summarized and discussed the work done in the field. Discussion can be improved by highlighting potential limitations and future challenges. I would also recommend changing the heading of ‘discussion’. It would be great to add Conclusion/summary at the end of this review.

Author Response

Response to Reviewers 1 and 2:

We thank the reviewers for their comments. As a response to the reviewers, we have now renamed the Discussion as “Conclusions and Perspectives” and has been shortened and adapted to avoid being vague and repeating the previous section. 

Potential limitations and future challenges have been highlighted. A brief note about other PTM examples in plant defence (such as acetylation) has been added. "The framework of plant defence" section has been shortened. The SnRK1/TOR section has been shortened.

Reviewer 2 Report

This review is written in a very interesting point of view that manifest the importance of PTMs in plant resistance processes. Authors make an extensive work based on the most well-studied PTMs such as phosphorylation, ubiquitination and SUMOylation. They have also included  TOR/SnRK1 interaction and how they are involved in the regulation of PTMs in different proteins related to plant defense. I really think that this is a compelling review that is an excellent resource of information for people working on plant immunity and pathogen interaction. 

I only have one recommendation to make and it is that the introductory parts of the review are somehow long especially the first part "the framework of plant defense" and also for the TOR/SnRK1 part. 

Overall, I still think that this is a high quality review that deserves to be published soon.
There are a couple of things that I may change though (but no major things).
1. In my opinion Figure 2 is very general and maybe it is not necessary to include.
2.The discussion is in some points a little vague (ex. "This work OFFERS potential approaches to exploit PTMs for crop improvement...) and also redundant with previous sections (ex.  "Exploiting PTMs to produce disease resistant crops"). I do recommend them to shorten the length of the discussion.
As a suggestion I would recommend including some data of other PTMs related to plant defense in the discussion at least mention further implications that may have on the signaling of biotic stress. 

Author Response

(The authors gave the same response as above.)
